# Constraints on nonlocality in networks from no-signaling and independence

Nicolas Gisin [1✉], Jean-Daniel Bancal [1✉], Yu Cai [1✉], Patrick Remy[1], Armin Tavakoli[1], Emmanuel Zambrini Cruzeiro[1,2], Sandu Popescu[3,4] & Nicolas Brunner[1]

The possibility of Bell inequality violations in quantum theory had a profound impact on our understanding of the correlations that can be shared by distant parties. Generalizing the concept of Bell nonlocality to networks leads to novel forms of correlations, the characterization of which is, however, challenging. Here, we investigate constraints on correlations in networks under the natural assumptions of no-signaling and independence of the sources. We consider the triangle network with binary outputs, and derive strong constraints on correlations even though the parties receive no input, i.e., each party performs a fixed measurement. We show that some of these constraints are tight, by constructing explicit local models (i.e. where sources distribute classical variables) that can saturate them. However, we also observe that other constraints can apparently not be saturated by local models, which opens the possibility of having nonlocal (but non-signaling) correlations in the triangle network with binary outputs.

[1] Département de Physique Appliquée, Université de Genève, Genève, Switzerland. [2] Laboratoire d'Information Quantique (LIQ), Université Libre de Bruxelles, Bruxelles, Belgium. [3] H.H. Wills Physics Laboratory, Tyndall Avenue, BS8 1TL Bristol, UK. [4] Institute for Theoretical Studies, ETH, Zurich, Switzerland. ✉email: Nicolas.Gisin@unige.ch; Jean-Daniel.Bancal@unige.ch; Yu.Cai@unige.ch

The no-signaling principle states that instantaneous communication at a distance is impossible. This imposes constraints on the possible correlations between distant observers. Consider the so-called Bell scenario[1], where each party performs different local measurements on a shared physical resource distributed by a single common source. In this case, the no-signaling principle implies that the choice of measurement (the input) of one party cannot influence the measurement statistics observed by the other parties (their outputs). In other words, the marginal probability distribution of each party (or subset of parties) must be independent of the input of any other party. These are the well-known no-signaling conditions, which represent the weakest conditions that correlations must satisfy in any reasonable physical theory[2], in the sense of being compatible with relativity. More generally, the no-signaling principle ensures that the information cannot be transmitted without any physical carrier. This provides a useful framework to investigate quantum correlations (which obviously satisfy the no-signaling conditions, but do not saturate them in general[2]) within a larger set of physical theories satisfying no-signaling; see e.g., refs. [2–9].

Recently, the concept of Bell nonlocality has been generalized to networks, where separated sources distribute physical resources to subsets of distant parties (Fig. 1). Assuming the sources to be independent from each other[10,11], arguably a natural assumption in this context, leads to many novel effects. Notably, it becomes now possible to demonstrate quantum nonlocality without the use of measurement inputs[11–15], but only by considering the output statistics of fixed measurements. Just recently, a first example of such nonlocality genuine to networks was proposed[15,16]. This radically departs from the standard setting of Bell nonlocality, and opens many novel questions. Characterizing correlations in networks (local or quantum) is however still very challenging at the moment, despite recent progress[17–28].

Moving beyond quantum correlations, this naturally raises the question of finding the limits of possible correlations in networks, assuming only no-signaling and independence (NSI) of the sources[22,29–33]. Here, we investigate this question and derive limits on correlations, which we refer to as NSI constraints. While our approach can in principle be applied to any network, we focus here on the well-known triangle network with binary outputs and no inputs, for which we obtain strong, and even tight NSI constraints. Specifically, we show that, despite the absence of an input, some statistics imply the possibility for one party to signal to others by locally changing (or not changing) the structure of the network. Formally, this amounts to considering a specific class of so-called network inflations, as introduced in ref. 22, which we show can lead to general and strong NSI constraints. Moreover, we prove that some of our NSI constraints are in fact tight, by showing that they can be saturated by correlations from explicit trilocal models, in which the sources distribute classical variables. Interestingly, however, it appears that not all of our NSI constraints can be saturated by trilocal models, which opens the possibility of having nonlocal (but nevertheless non-signaling) correlations in the triangle network with binary outputs. Finally, we conclude with a list of open questions.

## Results

**NSI constraints**. The triangle network (sketched in Fig. 1a) features three observers: Alice, Bob, and Charlie. Every pair of observers is connected by a (bipartite) source, providing a shared physical system. Importantly, the three sources are assumed to be independent from each other. Hence, the three observers share no common (i.e., tripartite) piece of information. Based on the received physical resources, each observer provides an output ($a$, $b$, and $c$, respectively). Note that the observers receive no input

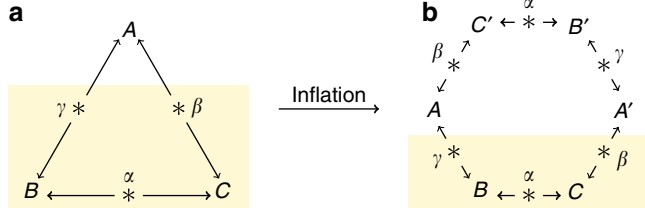

**Fig. 1 Inflation of the triangle network to the hexagon network.** In order to capture NSI constraints in the triangle network **a**, we consider an inflation to the hexagon network **b**. Importantly, from the point of view of Bob and Charlie, the two situations must be indistinguishable. If not, then Alice could (instantaneously) signal to Bob and Charlie, simply by locally modifying the network structure.

in this setting, contrary to standard Bell nonlocality tests. The statistics of the experiment are thus given by the joint probability distribution $p(a, b, c)$. We focus on the case of binary outputs: $a$, $b$, $c \in \{+1, -1\}$. It is then convenient to express the joint distribution as follows:

$$p(a, b, c) = \frac{1}{8}(1 + aE_A + bE_B + cE_C + abE_{AB} + acE_{AC} + bcE_{BC} + abcE_{ABC}), \quad (1)$$

where $E_A$, $E_B$, and $E_C$ are the single-party marginals, $E_{AB}$, $E_{BC}$, and $E_{AC}$ are the two-party marginals, and $E_{ABC}$ is the three-body correlator. Note that the positivity of $p(a, b, c)$ implies constraints on marginals, in particular $p(+++) + p(---) \geq 0$ implies

$$E_{AB} + E_{AC} + E_{BC} \geq -1 . \quad (2)$$

In the following, we will derive nontrivial constraints bounding and relating the single-party and two-party marginals of $p(a, b, c)$ under the assumption of NSI. While it seems a priori astonishing that the no-signaling principle can impose constraints in a Bell scenario, featuring no inputs for the parties, we will see that this is nevertheless the case in the triangle network.

The main idea is the following. Although one party (say Alice) receives no input, she could still potentially signal to Bob and Charlie by locally modifying the structure of the network. To see this, consider the hexagon network depicted in Fig. 1b, and focus on parties Bob and Charlie. From their point of view, the two networks (triangle and hexagon) should be indistinguishable. This is because all the modification required to bring the triangle network to the hexagon (e.g., by having Alice adding extra parties and sources) occurs on Alice's side, and can therefore be space-like separated from Bob and Charlie. If Alice, by deciding which network to use, could remotely influence the statistics of Bob and Charlie, this would clearly lead to signaling. Hence, we conclude that the local statistics of Bob and Charlie (i.e., the single-party marginals $E_B$ and $E_C$, as well as the two-party marginals $E_{BC}$) must be the same in the triangle and in the hexagon. To see that this condition really captures the possibility to signal, we could imagine a thought experiment in which we would give an input to Alice, which determines whether she modifies her network structure or not. If she does so and this has an incidence on the $E_{BC}$ marginal, then Bob and Charlie can learn about Alice's input, hence breaking the usual notion no-signaling condition. Note that the input considered here is however purely fictional, Alice's input is not present in the actual experiment.

From the above reasoning, we conclude that the joint output probability distribution for the hexagon, i.e., $p(a, b, c, a', b', c')$, must satisfy several constraints. In particular, one should have that

$$\sum b \, p(a, b, c, a', b', c') = \sum b' \, p(a, b, c, a', b', c') = E_B \quad (3)$$

$$\sum c\, p(a,b,c,a',b',c') = \sum c'\, p(a,b,c,a',b',c') = E_C \quad (4)$$

$$\sum bc\, p(a,b,c,a',b',c') = \sum b'c'\, p(a,b,c,a',b',c') = E_{BC}, \quad (5)$$

where all sums go over all outputs $a, b, c, a', b', c'$. From the independence of the sources, we obtain additional constraints, namely

$$\sum bb'\, p(a,b,c,a',b',c') = E_B^2 \quad (6)$$

$$\sum cc'\, p(a,b,c,a',b',c') = E_C^2 \quad (7)$$

$$\sum bb'c\, p(a,b,c,a',b',c') = E_{BC}E_B \quad (8)$$

$$\sum bcc'\, p(a,b,c,a',b',c') = E_{BC}E_C \quad (9)$$

$$\sum bcb'c'\, p(a,b,c,a',b',c') = E_{BC}^2 \; . \quad (10)$$

Clearly, we also get similar constraints when considering signaling between any other party (Bob or Charlie) to the remaining two.

Altogether, we see that NSI imposes many constraints on $p(a,b,c,a',b',c')$. Obviously, we also require that

$$p(a,b,c,a',b',c') \geq 0 \quad \text{and} \quad \sum p(a,b,c,a',b',c') = 1 \; . \quad (11)$$

Now reversing the argument, we see that the non-negativity of $p(a,b,c,a',b',c')$ imposes nontrivial constraints relating the single- and two-party marginals of the triangle distribution $p(a,b,c)$. To illustrate this, let us proceed with an example in a slightly simplified scenario, assuming all single-party marginals to be uniformly random, i.e., $E_A = E_B = E_C = 0$. In this case, we obtain

$$
\begin{aligned}
64\, p(a,b,c,a',b',c') =\; & 1 + (ab + a'b')E_{AB} + (bc + b'c')E_{BC} + (ca' + c'a)E_{AC} \\
& + (abc + a'b'c')F_3 + (bca' + b'c'a)F_3' + (ca'b' + c'ab)F_3'' \\
& + aa'bb'E_{AB}^2 + bb'cc'E_{BC}^2 + aa'cc'E_{AC}^2 + aa'(bc + b'c')F_4 \\
& + bb'(ca' + c'a)F_4' + cc'(ab + a'b')F_4'' + aa'bb'(c + c')F_5 \\
& + bb'cc'(a + a')F_5' + aa'cc'(b + b')F_5'' + aa'bb'cc'F_6 \geq 0
\end{aligned}
\quad (12)
$$

Importantly, notice that the above expression contains a number of variables (of the form $F_X$) that are uncharacterized; these represent X-party correlators in the hexagon network, see Supplementary Note 1 for more details. Hence, we obtain a set of inequalities imposing constraints on our variables of interest (i.e., $E_{AB}$, $E_{BC}$, and $E_{AC}$), but containing also additional variables that we would like to discard. This can be done systematically via the algorithm of Fourier–Motzkin elimination[34]. Note that here we need to treat the squared terms, such as $E_{AB}^2$, as new variables, independent from $E_{AB}$, so that we get a system of linear inequalities. Solving the latter, and taking into account positivity constraints as in Eq. (2), we obtain a complete characterization of the set of two-body marginals (i.e., $E_{AB}$, $E_{BC}$, and $E_{AC}$) that are compatible with NSI in the triangle network (for a hexagon inflation and uniform single-party marginals), in terms of a single inequality

$$(1 - E_{AB})^2 - E_{BC}^2 - E_{AC}^2 \geq 0 \; , \quad (13)$$

and its symmetries (under relabeling of the parties and of the outputs). This implies a more symmetric, but slightly weaker inequality:

$$(1 + E_{AB})^2 + (1 + E_{BC})^2 + (1 + E_{AC})^2 \leq 6 \; . \quad (14)$$

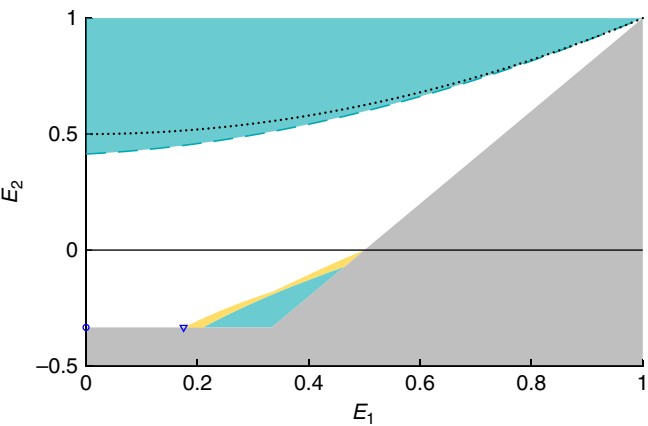

**Fig. 2 Region of allowed correlations for symmetric distributions; projection in the plane $E_2$ vs $E_1$.** The turquoise region is ruled out by NSI constraints, while the gray region is excluded from simple positivity constraints. The white region is accessible via trilocal models. Correlations in the yellow region satisfy NSI constraints (from the hexagon inflation), but we could not find a trilocal model for them. The constraint Eq. (34) of ref. [22] is shown in dotted black. The dashed turquoise curve corresponds to the NSI inequality Eq. (15), which turns out to be tight. Explicit trilocal models are also obtained for the correlations marked by blue dots (Supplementary Note 2).

Note that when $E_{AB} = E_{BC} = E_{AC} \equiv E_2$, we get simply $E_2 \leq \sqrt{2} - 1 \approx 0.41$.

Next, we consider the symmetric case (i.e., $E_A = E_B = E_C \equiv E_1$ and $E_{AB} = E_{BC} = E_{AC} \equiv E_2$) and obtain nontrivial NSI constraints on the possible values of $E_1$ and $E_2$ (Fig. 2). In particular, correlations compatible with NSI must satisfy the following inequality

$$(1 + 2|E_1| + E_2)^2 \leq 2(1 + |E_1|)^3 \; . \quad (15)$$

Let us move now to the most general case, with arbitrary values for single- and two-party marginals. For a given set of values $E_A$, $E_B$, $E_C$, $E_{AB}$, $E_{BC}$, and $E_{AC}$, it is possible here to determine via a linear program whether this set is compatible with NSI or not (Supplementary Note 1). More generally, obtaining a characterization of the NSI constraints in terms of explicit inequalities (as above) is challenging, due mainly to the number of parameters and nonlinear constraints. We nevertheless obtain that the following inequality represents an NSI constraint

$$
\begin{aligned}
& (1 + |E_A| + |E_B| + E_{AB})^2 \\
& + (1 + |E_A| + |E_C| + E_{AC})^2 \\
& + (1 + |E_B| + |E_C| + E_{BC})^2 \\
& \leq 6(1 + |E_A|)(1 + |E_B|)(1 + |E_C|) \; .
\end{aligned}
\quad (16)
$$

A proof of this general inequality is given in Supplementary Note 1. Note that this inequality reduces to Eq. (14) when $E_A = E_B = E_C = 0$, as well as to Eq. (15) for the symmetric case.

It is worthwhile discussing the connection between our approach and the inflation technique presented in refs. [22,25]. There, the main focus is on using inflated networks for deriving constraints on correlations achievable, with classical resources. In that case, information can be readily copied, so that sources can send the same information to several parties. Ultimately, this allows for a full characterization of correlations achievable with classical resources[22]. Copying information is however not possible in our case, as no-signaling resources cannot be perfectly cloned in general[6]. Hence only inflated networks with bipartite

sources can be considered in our case, such as the hexagon. A discussion of these ideas can be found in Section V.D of ref. [22], where the idea of using inflation to limit no-signaling correlations in networks is mentioned. Here, we derive explicitly bounds that all correlations satisfying the NSI constraints, whether quantum of post-quantum, have to satisfy, and identify the physical principle behind them.

Finally, the choice of the hexagon inflation deserves a few words. As seen from Fig. 1b, it is judicious to consider inflated networks forming a ring, with a number of parties that is a multiple of three. Intuitively, this should enforce the strongest constraints on the correlations of the inflated network; in particular, all single- and two-body marginals are fixed by the correlations of the triangle. This would not be the case when considering inflations to ring networks, with a number of parties that is not divisible by three.

**Tightness**. A natural question is whether the constraints we derived above, that are necessary to satisfy NSI, are also sufficient. There is a priori no reason why this should be the case. Of course, starting from the triangle network, there are many (in fact infinitely many) possible extended networks that can be considered, and no-signaling must be enforced in all cases. For instance, instead of extending the network to a hexagon (as in Fig. 1), Alice could consider an extension to a ring network featuring 9, 12, or more parties. Clearly, such extensions could lead to stronger constraints than those derived here for the hexagon network.

Nevertheless, we show that some of the constraints we obtain above are in fact tight, i.e., necessary and sufficient for NSI. We prove this by presenting explicit correlations (constructed within a generalized probabilitic theory satisfying NSI) that saturate these constraints. In fact, we consider simply the case where all sources distribute classical variables to each party, which we refer to as trilocal models. The latter give rise to correlations of the form

$$p(a, b, c) = \int \mu(\alpha)d\alpha \int \nu(\beta)d\beta \int \omega(\gamma)d\gamma \\ p_A(a|\beta, \gamma) \; p_B(b|\alpha, \gamma) \; p_C(c|\alpha, \beta),$$ (17)

where $\alpha$, $\beta$, and $\gamma$ represent the three local variables distributed by each source, with arbitrary probability densities $\mu(\alpha)$, $\nu(\beta)$, and $\omega(\gamma)$. Also, $p_A(a|\beta, \gamma)$ represents an arbitrary response function for Alice, and similarly for $p_B(b|\alpha, \gamma)$ and $p_C(c|\alpha, \beta)$. Note that such trilocal models represents a natural extension of the concept of Bell locality to networks (see e.g., refs. [10,19]).

We first consider the case of symmetric distributions, i.e., characterized by the two parameters $E_1$ and $E_2$, and seek to determine the set of correlations that can be achieved with trilocal models. As shown in Fig. 2, it turns out that almost all NSI constraints can be saturated in this case, in particular the inequality (15). After performing a numerical search, we could construct explicitly some of these trilocal models, which involve up to ternary local variables (see Supplementary Note 2 for details). Moreover, we compare our NSI constraint (15) to the one derived in ref. [22] (see Eq. (34)), and find that the present one is stronger, and in fact tight (Fig. 2). Note also that a previous work derived an NSI constraint based on entropic quantities[29]; such constraints are however known to be generally weak, as entropies are a coarse-graining of the statistics, which no longer distinguishes between correlations and anticorrelations.

As seen from Fig. 2, there is however a small region (in yellow) that is compatible with NSI (considering the hexagon inflation), but for which we could not construct a trilocal model. Whether this gap can be closed by considering more sophisticated local models (using variables of larger alphabet) or whether stronger no-signaling bounds can be obtained is an interesting open question.

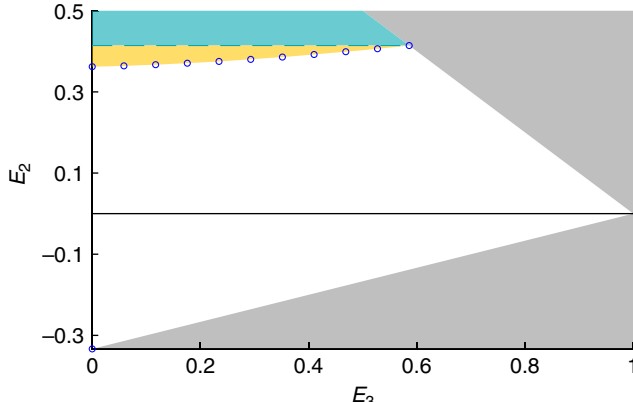

**Fig. 3 Region of allowed correlations for symmetric distributions with $E_1 = 0$; represented in the plane $E_2$ vs $E_3$.** The turquoise region is ruled out by NSI constraints (dashed turquoise line given by Eq. (15)), while the gray region is excluded from simple positivity constraints. The white region is accessible via trilocal models. Correlations in the yellow region satisfy NSI constraints (from the hexagon inflation), but we could not find a trilocal model for them. Explicit trilocal models are also obtained for the correlations marked by blue dots (see Supplementary Note 2).

For the triangle network with binary outcomes, any trilocal distribution can be obtained by considering shared variables of dimension (at most) six, and deterministic response functions[24].

In fact, another (and arguably much more interesting) possibility would be that this gap cannot be closed, as it would feature correlations with binary outcomes satisfying NSI, but that are nevertheless non-trilocal. To further explore this question, let us now focus on the case where single-party marginals vanish, i.e., $E_1 = 0$. We investigate the relation between two-party marginals $E_2$ and the three-party correlator $E_3 = E_{ABC}$, comparing NSI constraints and trilocal models. Notice that the NSI constraints we obtain here do not involve $E_3$ (as the latter cannot be recovered within the analysis of the hexagon). Hence NSI imposes only $E_2 \leq \sqrt{2} - 1$, while positivity of $p(a, b, c)$ imposes other constraints. This is shown in Fig. 3, where we also seek to characterize the set of correlations achievable via trilocal models (proceeding as above). Interestingly, we find again a potential gap between trilocal correlations and NSI constraints. This should however be considered with care. First, the NSI constraints obtained from the hexagon may not be optimal (see Discussion section). Second, there could exist more sophisticated trilocal models (e.g., involving higher-dimensional variables) that could lead to a stronger correlations (i.e., cover a larger region in Fig. 3). Note also that we investigated whether quantum distributions satisfying the independence assumption exist outside of the trilocal region, but we could not find any example (we performed a numerical search, considering entangled states of dimension up to $4 \times 4$).

Finally, note that we also performed a similar analysis for the case where single-party marginals vanish, but two-body marginals are not assumed to be identical to each other. Here, we find that inequality (13) can be saturated in a few specific cases. However, there also exist correlations satisfying the NSI bounds that do not seem to admit a trilocal model; details in Supplementary Note 1.

## Discussion

We discussed the constraints arising on correlations in networks, under the assumption of NSI of the sources. We focused our attention on the triangle network with binary outputs for which we derived strong constraints, including tight ones. Our work raises a number of open questions that we now discuss further.

A first question is whether the constraints we derive (necessary under NSI), could also be sufficient. We believe this not to be the case, as stronger NSI constraints could arise from inflations of the triangle to more complex networks (e.g., loop networks with an arbitrary number of parties). Note that there could also exist different forms of no-signaling constraints, that cannot be enforced via inflation. In this respect, we compare in Supplementary Note 1 our NSI constraints with the recent work of ref. [32] proposing a very different approach to this problem, using the Finner inequality. A notable difference is that the latter imposes constraints on tripartite correlations, which is not the case here.

Another important question is whether there could exist nonlocality in the simplest triangle network with binary outcomes. That is, can we find a $p(a, b, c)$ that satisfies NSI, but that is nevertheless non-trilocal? While we identified certain potential candidate distributions for this, we could not prove any conclusive result at this point. We cannot exclude the possibilities that (i) these correlations are in fact not compatible with NSI (as there exist stronger NSI constraints) or (ii) these correlations can in fact be reproduced by a trilocal model. In order to address point (i), one could try to reproduce these correlations via an explicit NSI model, for instance considering that all sources emit no-signaling resources (such as nonlocal boxes[2]) which could then be wired together by the parties. To address point (ii), one could show that these correlations violate a multilocality inequality for the triangle network. Of course finding such inequalities is notably challenging, see e.g., ref. [13].

Furthermore, it would be interesting to derive NSI constraints for other types of networks. Indeed, the approach developed here can be straightforwardly used. Cases of high interest are general loop networks, as well as the triangle network with larger output alphabet (where examples of quantum nonlocality are proven to exist[11,15]).

Finally, a more fundamental question is whether any correlation satisfying the complete NSI constraints can be realized within an explicit physical theory satisfying no-signaling (the latter are usually referred to as generalized probabilistic theories[6]). While this is the case in the standard Bell scenario (where all parties share a common resource), it is not clear if that would also be the case in the network scenario.

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

## Acknowledgements

We thank Stefano Pironio, Marc-Olivier Renou, Denis Rosset, and Elie Wolfe for discussions. We acknowledge financial support from the Swiss national science foundation (Starting grant DIAQ, NCCR-QSIT, and NCCR-Swissmap). E.Z.C. acknowledges support by the Swiss National Science Foundation via the Mobility Fellowship P2GEP2_188276.

## Author contributions

N.G., S.P., and N.B. came up with the idea of the method. N.G., J.-D.B., Y.C., P.R., A.T., E.Z.C., S.P., and N.B. participated in deriving the results, and writing and editing the manuscript.

## Competing interests

The authors declare no competing interests.
