## [Peer Review File · Nature Communications]

Reviewers' Comments:

Reviewer #1:

Remarks to the Author:

Note: I uploaded a nicely LaTeX-typeset version of this report as "(Optional) supplementary material relevant to [my] review". Please see Review Attachment #1. There is no advantage in reading this ugly text-only version, but I include it in this box just in case my supplementary material does not go through for any reason whatsoever. Here goes...

This is a truly remarkable manuscript, which goes cleverly and squarely against the conventional wisdom according to which "the no-signaling principle can*not* impose constraints in a Bell scenario featuring no inputs for the parties", where I quote from page 2 of the paper, having only added the italicized "*not*". The paper rightfully says that this "seems a priori astonishing". The web interface for submitting referee reports asks in particular if I "feel that the paper will influence thinking in the field". My answer is a resounding yes, precisely because it goes against conventional wisdom, which I think is the only approach to making fundamental (as opposed to incremental) progress in science. Furthermore, the paper comes with fascinating open questions. Consequently, I recommend enthusiastically that the paper be accepted for publication in *Nature Communications*.

At this point, I must make the disclaimer that I did not thoroughly verify all the math in this paper. Actually, I gave up trying to follow the math in detail when I reached scary Equation (4)! Nevertheless, I have great confidence in the authors' mathematical abilities and I am therefore confident in the mathematical correctness of this paper. Also, I must admit that I did not read the Supplementary Information as thoroughly as the Main Text, but I read it nevertheless.

In addition to the disclaimer above, which reflects nothing but my own shortcomings, I have some reservations about the paper itself, which I detail below. This referee report contains the more important remarks. In addition, I provide an annotated version of the authors' PDF file as Review Attachment #2, in which I provide less important comments that should nevertheless be considered by the authors, albeit optionally.

As I stated already, this paper breaks new ground in exploring the possibility of exploiting the no-signalling principle in Bell scenarios featuring no inputs for the parties. However, it is not clear what is novel here compared to other papers published previously this year by some of the same authors (e.g. [13,15]) or competing authors (e.g. [17]). Note that Ref. [13] was selected by *Entropy* as a "feature paper" whereas Ref. [15] was selected by the American Physical Society as an "Editor's Suggestion" as well as "Featured in Physics", which vindicates the importance I see in this novel and surprising research field.

It may be that a specificity of the current paper is that it considers the more restricted (hence more difficult) scenario in which all outputs are binary. Note that the authors do not mention before page 2 that they "focus on the case of binary outputs". Maybe this should be stated as early as the Abstract and possibly even in the title, especially if this is indeed a fundamental ingredient that sets this work apart from previous work. Perhaps another originality of the current paper is that the trilocal models that are used to saturate NSI bounds are powered by *classical* variables, whereas the former work with which I am familiar considered sources that distribute quantum resources. If this is indeed the case, perhaps it should be explicitly stated that this is a specificity of this research.

Another reservation is that the no-signalling principle is harnessed by reasoning on the possibility that Alice could signal to other parties (but should not) by changing or not her local topology of the network (for instance from triangular to hexagonal). This is very clever, but how is this choice fundamentally different from choosing to perform one measurement or another in a standard Bell scenario (i.e. a scenario in which parties receive inputs that determine which measurement to

perform)?

A slightly more technical point occurs on page 4, right column, starting at line 15: "Notice that the NSI constraints do not involve E_3 (as the latter cannot be recovered within the analysis of the hexagon)". I agree that the inflation technique used in this paper to derive NSI constraints cannot impose restrictions on E_3 . However, is it totally impossible that a completely *different* technique might be used to derive NSI constraints that involve E_3 ? Perhaps the quoted statement is too sweeping and should be qualified.

Even more technical is Equation (3) in the Supplementary Information, where we consider all F such that some condition holds and then take the maximum over all these F of... 1! You have to go and see Equation (3) to understand what I mean.

Less importantly, I noticed that references [15] and [16] are identical!

In summary, I highly recommend that this paper be accepted despite my reservations above. In addition to this report, please refer to my enclosed annotated PDF of the paper (including the Supplementary Information), which contains several less important comments in addition to repeating some---but not all---of the above.

Gilles Brassard FRS

Reviewer #2:

Remarks to the Author:

This is great work. The authors found an ingenious way to apply the inflation technique to the triangle scenario in order to bound not only classical, but quantum and supra-quantum correlations. With their technique they developed non-trivial inequalities to bound the correlations, and identified candidates for dichotomic distributions that might be quantum but not trilocal.

I am, however, concerned that the title and the abstract do not reflect accurately the contents of the manuscript. The title claims that the constraints come from no-signalling. But the informal ideal of no-signalling, as described in the the beginning of the right column of page 2, would lead to constraints of the form

$$\sum_{a,b,c} b p(a,b,c) = \sum_{a,b,c,a',b',c'} b p(a,b,c,a',b',c')$$

which is clearly trivial. The constraints the authors actually use, in the unnumbered equation (please number all your equations!) before equation (3), are just identifying B with B' , C with C' , and so on, as is standard in the inflation technique. (the constraints from equation (3) are in fact independence constraints, as claimed by the authors).

As for the abstract, in its last sentence the authors say "However, we also observe that other constraints can apparently not be saturated by local models, which opens the possibility of having nonlocal (but non-signaling) correlations in the triangle network." which is a bizarre statement, since it has been known for years that there are nonlocal (but non-signalling) correlations in the triangle scenario. It is also inconceivable that the authors are not aware of this, since some of these correlations have been found by a subset of the authors of this manuscript. The problem that is open is not whether these correlations exist at all, but whether dichotomic correlations do. This must be stated in the abstract, as otherwise the manuscript as a whole becomes quite confusing. Note that this same bizarre sentence is repeated at the end of the introduction.

I think the manuscript would benefit a lot from a discussion of the case of 4 outputs per party. Clearly the author's technique also applies there (even though the resulting inequalities will

probably look nasty), and since nonlocal correlations are known in this case, it would provide a good illustration of the capabilities of the technique.

Some minor issues follow:

1 - after equation (4): why do you eliminate $E_{\{ABC\}}$? It is clearly the inequality (in the term abc_{F_3}), and you can't say it is uncharacterised. Doesn't it give any useful constraint? Also the F_X variables are not defined, one has to guess what they mean from the notation. You should at least mention in the supplementary note that their definition is indeed what one would guess (if this is indeed the case)

2 - equation (8) is illegible, and it is not clear what $\underset{ABC}{\circlearrowleft}$. Since this is a key conjecture of the manuscript, it would be worth it to write down this inequality explicitly.

3 - This sentence in the end of section two is misleading:

"In fact, another (and arguably much more interesting) possibility would be that this gap cannot be closed, as it would feature correlations satisfying NSI but that are nevertheless non-trilocal."

Again, it is well-known that such correlation exist (except in the dichotomic case), but the sentence suggests otherwise.

4 - In the sentence "First, the NSI constraints obtained from the hexagon may not be optimal (see conclusion)." maybe the authors mean "discussion", instead of "conclusion", as there is no conclusion.

5 - In the sentence "Here we find that inequality (5) can be saturated in few specific cases.", maybe the authors mean instead "Here we find that inequality (5) can be saturated in a few specific cases."

Reviewer #3:

Remarks to the Author:

The paper considers the implications of independence and non-signalling for the triangle network. The authors derive correlator inequalities that are valid when imposing restrictions on some of the correlators (setting some correlators to zero or requiring several of them to be equal), a generalisation of the inequalities that lifts these additional limitations is conjectured. Some of the inequalities are shown to be tight in the sense that they can be saturated with classical models in the triangle network.

I like the idea of generalising ideas inspired by graph inflation and applying them to arbitrary non-signalling theories. I think, however, that the present paper does not make sufficient progress on this to deserve publication in Nature Communications for the following reasons. The idea of considering correlations in arbitrary networks restricted by non-signalling and independent sources is not new (see for instance [30], where a theory-independent constraint for the triangle network was derived). Similarly, it is already known that the triangle network features nonlocal but non-signalling correlations (as the authors mention, this can for instance be found in [16]). While I do believe that the method the authors introduce here may be of interest for obtaining useful constraints on correlations in various networks, I believe that the authors have not presented strong enough results to back this up: the constraints that are proved in the present paper are only valid under additional restrictions on the correlators and computational challenges seem to have so far prevented a generalisation.

The paper is nicely written and I am convinced of its correctness. I have several specific comments, where I think the manuscript could be improved.

1) All inequalities that are derived in the present paper are restricted to special cases. A comment on the robustness of these in cases where the marginals are only approximately uniform or the symmetries only approximately met would therefore be relevant.

2) I am a bit confused by the claim ``Here, for the first time, we derive explicitly bounds that all correlations satisfying the NSI constraints, whether quantum or post-quantum, have to satisfy.' at the end of page 3. It is mentioned in Section 5.4 of [23] that inflations without inflationary fan-outs apply to all GPTs. These include the so-called Cut Inflation, for which Example 4 of [23] provides an inequality for correlators. A clarification why this is not considered to be such an example and a comparison of the inequalities of the present manuscript to that of Example 4 of [23] would be helpful here.

3) There is further recent work that proposed theory-independent constraints for networks (I know of [23], [30], [Chaves and Budroni, PRL 116, 240501, 2016] and [Weilenmann and Colbeck, ArXiv:1812.04327]). A comment on the advantages of the present work over these attempts would be of interest to the community.

4) There seem to be two typos concerning the referencing. [15] and [16] seem to refer to the same paper. In the Supplementary Information, some of the references point to equations rather than figures.

Response to Reviewer 1

- *This is a truly remarkable manuscript, which goes cleverly and squarely against the conventional wisdom according to which "the no-signaling principle cannot impose constraints in a Bell scenario featuring no inputs for the parties", where I quote from page 2 of the paper, having only added the italicized "not". The paper rightfully says that this "seems a priori astonishing". The web interface for submitting referee reports asks in particular if I "feel that the paper will influence thinking in the field". My answer is a resounding yes, precisely because it goes against conventional wisdom, which I think is the only approach to making fundamental (as opposed to incremental) progress in science. Furthermore, the paper comes with fascinating open questions. Consequently, I recommend enthusiastically that the paper be accepted for publication in Nature Communications.*

We thank the Referee for the time and effort he invested in reading and evaluating our manuscript. We are glad to see his appreciation of the fundamental value of our findings and that he enthusiastically supports its publication in Nature Communications.

- *At this point, I must make the disclaimer that I did not thoroughly verify all the math in this paper. Actually, I gave up trying to follow the math in detail when I reached scary Equation (4)! Nevertheless, I have great confidence in the authors' mathematical abilities and I am therefore confident in the mathematical correctness of this paper. Also, I must admit that I did not read the Supplementary Information as thoroughly as the Main Text, but I read it nevertheless.*

In addition to the disclaimer above, which reflects nothing but my own shortcomings, I have some reservations about the paper itself, which I detail below. This referee report contains the more important remarks. In addition, I provide an annotated version of the authors' PDF file as Review Attachment #2, in which I provide less important comments that should nevertheless be considered by the authors, albeit optionally.

We thank the Referee for the constructive comments provided on the annotated PDF file. We have taken them into account to improve the manuscript.

- *As I stated already, this paper breaks new ground in exploring the possibility of exploiting the no-signalling principle in Bell scenarios featuring no inputs for the parties. However, it is not clear what is novel here compared to other papers published previously this year by some of the same authors (e.g. [13,15]) or competing authors (e.g. [17]). Note that Ref. [13] was selected by Entropy as a "feature paper" whereas Ref. [15] was selected by the American Physical Society as an "Editor's Suggestion" as well as "Featured in Physics", which vindicates the importance I see in this novel and surprising research field.*

It may be that a specificity of the current paper is that it considers the more restricted (hence more difficult) scenario in which all outputs are binary. Note that the authors do not mention before page 2 that they "focus on the case of binary outputs". Maybe this should be stated as early as the Abstract and possibly even in the title, especially if this is indeed a fundamental ingredient that sets this work apart from previous work. Perhaps another originality of the current paper is that the trilocal models that are used to saturate NSI bounds are powered by classical variables, whereas the former work with which I am familiar considered sources that distribute quantum resources. If this is indeed the case, perhaps it should be explicitly stated that this is a specificity of this research.

We agree with the Referee additional explanations regarding how the current work relates to Ref. [13,15,17] would be useful. The main difference is that these works restrict their attention to the characterization of quantum correlations in the triangle scenario. Here, we are interested in describing the constraints that must be satisfied by all no-signalling theories. We modified the introduction in a few key places to clarify this point. For instance, the beginning of the third paragraph now starts with "Moving beyond quantum correlations, (...)". Note that Ref. [17]

discusses both the cases quantum and no-signaling correlations, but does not prove (only conjecture) a general NSI constraint in the form of the Finner inequality.

As the Referee points out, another difference with Refs [13,15] is the fact that our work considers the case of binary outputs. We now mention this right from the abstract.

- *Another reservation is that the no-signalling principle is harnessed by reasoning on the possibility that Alice could signal to other parties (but should not) by changing or not her local topology of the network (for instance from triangular to hexagonal). This is very clever, but how is this choice fundamentally different from choosing to perform one measurement or another in a standard Bell scenario (i.e. a scenario in which parties receive inputs that determine which measurement to perform)?*

Our construction can be framed in a scenario in which the parties receive inputs that determine which topology to use (instead of which measurement to perform). However, contrary to the usual case, these “inputs” are virtual: they are not present in the actual experiment, which is performed with fixed measurements (i.e. no inputs). In this sense it is different from the usual scenario. We now provide on page 2 an interpretation of our construction in terms of a thought experiment in which parties would be given inputs. Hopefully, this helps to relate what we present with the setting in which the non-signalling principle is usually described.

- *A slightly more technical point occurs on page 4, right column, starting at line 15: "Notice that the NSI constraints do not involve E_3 (as the latter cannot be recovered within the analysis of the hexagon)". I agree that the inflation technique used in this paper to derive NSI constraints cannot impose restrictions on E_3 . However, is it totally impossible that a completely different technique might be used to derive NSI constraints that involve E_3 ? Perhaps the quoted statement is too sweeping and should be qualified.*

We agree that it is the specific way of deriving NSI constraints used here which forbids the involvement of and E_3 term, i.e. nothing prevents a priori a NSI constraint from depending on E_3 . We now state explicitly that “the NSI constraints we obtain here do not involve E_3 ”.

- *Even more technical is Equation (3) in the Supplementary Information, where we consider all F such that some condition holds and then take the maximum over all these F of... 1! You have to go and see Equation (3) to understand what I mean.*

We agree that this notation might be puzzling at first. In fact it is a standard way of expressing feasibility problems: as a constrained optimization with trivial objective function. Generally speaking, the solution of a convex optimization is a set of value assignments for the variables. This assignment must satisfy the constraints and maximize the objective function. If the set of constraints admit at least one solution, this set is nonempty and allows to evaluate the value of the objective function (1 in the present case). If the constraints do not admit a solution, the set of solutions is empty and the problem is said to be ‘infeasible’. The value of the objective function is then of no importance, since the domain on which the function can be evaluated is empty. Whether the problem is feasible or not is precisely what we are testing here. We added an explanation for this notation next to Equation (3).

- *Less importantly, I noticed that references [15] and [16] are identical!*

We thank the Referee for pointing out this mistake, which escaped our attention. There is now only reference to this work.

- *In summary, I highly recommend that this paper be accepted despite my reservations above. In addition to this report, please refer to my enclosed annotated PDF of the paper (including the Supplementary Information), which contains several less important comments in addition to repeating some---but not all---of the above.*

Once again we thank the Referee for his constructive comments.

Response to Reviewer 2

- *This is great work. The authors found an ingenious way to apply the inflation technique to the triangle scenario in order to bound not only classical, but quantum and supra-quantum correlations. With their technique they developed non-trivial inequalities to bound the correlations, and identified candidates for dichotomic distributions that might be quantum but not trilocal.*

We thank the Referee for their time and effort invested in reading and evaluating our manuscript. We are glad to see they appreciate the ingenuity of our findings.

- *I am, however, concerned that the title and the abstract do not reflect accurately the contents of the manuscript. The title claims that the constraints come from no-signalling. But the informal ideal of no-signalling, as described in the the beginning of the right column of page 2, would lead to constraints of the form*

$$\sum_{\{a,b,c\}} b p(a,b,c) = \sum_{\{a,b,c,a',b',c'\}} b p(a,b,c,a',b',c')$$

which is clearly trivial. The constraints the authors actually use, in the unnumbered equation (please number all your equations!) before equation (3), are just identifying B with B', C with C', and so on, as is standard in the inflation technique. (the constraints from equation (3) are in fact independence constraints, as claimed by the authors).

No-signalling conditions are indeed generally expressed through equations of the form mentioned by the Referee. These equations express the fact that the marginal probability of some party should not depend on a choice of setting made by another party. Here, it is also this potential dependency of a marginal on a choice made by another party that we are testing. The difference, however, is the way in which the second party expresses his/her choice: instead of being through a measurement setting, as usually considered, it is done here through a choice of network configuration. Nevertheless, in both cases the critical question is whether the second party can signal to the first one.

In order to see this unambiguously, we can imagine a situation in which Alice receives an input $x=0,1$ that determines which topology of the network she should use (instead of which measurement to perform): if $x=0$, she uses the triangle configuration, and if $x=1$, she locally expands the triangle into an hexagon. In this hypothetical setting, we can ask whether the bipartite marginal E_{BC} depends on Alice's setting x or not. Mathematically, this amount to compare $E_{\{BC|x=0\}}$ and $E_{\{BC|x=1\}}$. Under the constraint of non-signalling, there should be no difference between the two marginals $E_{\{BC|x=0\}}$ and $E_{\{BC|x=1\}}$. This is both a non-signalling condition in the usual form, and precisely the condition that we require. Hence, the constraints we provide must be satisfied in

any physical theory that satisfies the non-signalling principle. The subtlety here being that the “input” x is virtual: it is not physically present in the actual triangle experiment.

We now provide on page 2 an interpretation of our construction in terms of a thought experiment in which parties would be given inputs. Hopefully, this helps to relate what we present with the setting in which the non-signalling principle is usually described.

We thank the Referee for pointing out the usefulness of numbering all equations. We now number all our equations.

- *As for the abstract, in its last sentence the authors say "However, we also observe that other constraints can apparently not be saturated by local models, which opens the possibility of having nonlocal (but non-signaling) correlations in the triangle network." which is a bizarre statement, since it has been known for years that there are nonlocal (but non-signalling) correlations in the triangle scenario. It is also inconceivable that the authors are not aware of this, since some of these correlations have been found by a subset of the authors of this manuscript. The problem that is open is not whether these correlations exist at all, but whether dichotomic correlations do. This must be stated in the abstract, as otherwise the manuscript as a whole becomes quite confusing. Note that this same bizarre sentence is repeated at the end of the introduction.*

We thank the Referee for this comment. Following the Referee’s suggestion, we now better highlight the fact that our results only require binary outputs, already in the abstract. In particular, the last sentence of the abstract has been modified to “opens the possibility of having nonlocal (but non-signalling) correlations in the triangle network with binary outputs.”

- *I think the manuscript would benefit a lot from a discussion of the case of 4 outputs per party. Clearly the author's technique also applies there (even though the resulting inequalities will probably look nasty), and since nonlocal correlations are known in this case, it would provide a good illustration of the capabilities of the technique.*

We agree with the Referee that the case of 4 outputs per party is an interesting one. However, we believe that it lies beyond the scope of the current work. The main message of our manuscript is that meaningful conditions can be obtained for network configurations under the simple conditions of non-signalling and independence. We believe that this is already demonstrated with the current discussion with 2 outputs. Moreover, as the Referee suggests, moving to the case of 4 outputs will considerably increase the complexity of the problem, and we could not so far find simple and elegant constraints. Indeed, in the case of binary outputs, the use of binary correlators greatly simplifies the derivation of constraints. Using generalized correlators is in principle possible, but complicates seriously the calculations. At this point, we believe that discussing these more complicated case would not help the reader to grasp the main message of our paper. This is nevertheless a very interesting problem for future work.

- *Some minor issues follow:*

- 1 - after equation (4): why do you eliminate $E_{\{ABC\}}$? It is clearly the inequality (in the term $abc F_3$), and you can't say it is uncharacterised. Doesn't it give any useful constraint? Also the F_X variables are not defined, one has to guess what they mean from the notation. You should at least mention in the supplementary note that their definition is indeed what one would guess (if this is indeed the case)*

The term $E_{\{ABC\}}$ refers to the tripartite correlation term that parties A, B and B observe in the triangle scenario (Fig. 1(a)). In contrast, the term F_3 refers to the tripartite term that A, B and C observe in the hexagon configuration (Fig. 1(b)). In Fig. 1(a), Alice, Bob and Charlie are linked

with three sources alpha, beta, and gamma, but in Fig. 1(b) Alice, Bob and Charlie are linked only with two sources alpha and gamma. These two terms thus don't need to be identical (in many cases they won't). Since F_3 does not appear in the triangle configuration (there are no three parties linked with only two sources there), it is a free variable. We thus need to eliminate it in order to obtain constraints involving only the physical terms that appear in the triangle scenario Fig. 1(a).

We thank the Referee for pointing out that the meaning of these E and F variables was not clearly introduced. We now explain the meaning of these variables in details in the Supplementary Note I.

- 2 - equation (8) is illegible, and it is not clear what $\underset{ABC}{\circlearrowleft}$. Since this is a key conjecture of the manuscript, it would be worth it to write down this inequality explicitly.

Equation (8) (now (10)) has been fully rewritten. It is now also proven analytically to be a valid NSI constraint.

- 3 - This sentence in the end of section two is misleading:

"In fact, another (and arguably much more interesting) possibility would be that this gap cannot be closed, as it would feature correlations satisfying NSI but that are nevertheless non-trilocal."

Again, it is well-known that such correlation exist (except in the dichotomic case), but the sentence suggests otherwise.

We thank the Referee for this feedback. To clarify this sentence, we now mention explicitly that we are talking about the dichotomic case here.

- 4 - In the sentence "First, the NSI constraints obtained from the hexagon may not be optimal (see conclusion)." maybe the authors mean "discussion", instead of "conclusion", as there is no conclusion.

Indeed this was a mistake. We now refer here to the discussion section, instead of the conclusion.

- 5 - In the sentence "Here we find that inequality (5) can be saturated in few specific cases.", maybe the authors mean instead "Here we find that inequality (5) can be saturated in a few specific cases."

We followed the Referee's advice.

Response to Reviewer 3

- The paper considers the implications of independence and non-signalling for the triangle network. The authors derive correlator inequalities that are valid when imposing restrictions on some of the correlators (setting some correlators to zero or requiring several of them to be equal), a generalisation of the inequalities that lifts these additional limitations is conjectured. Some of the inequalities are shown to be tight in the sense that they can be saturated with classical models in the triangle network. I like the idea of generalising ideas inspired by graph inflation and applying them to arbitrary non-signalling theories. I think, however, that the present paper does not make sufficient progress on this to deserve publication in Nature Communications for the following reasons. The idea of considering correlations in arbitrary networks restricted by non-signalling and independent sources is not new (see for instance [30], where a theory-independent constraint for the triangle network was derived). Similarly, it is already known that the triangle network features nonlocal but non-signalling correlations

(as the authors mention, this can for instance be found in [16]). While I do believe that the method the authors introduce here may be of interest for obtaining useful constraints on correlations in various networks, I believe that the authors have not presented strong enough results to back this up: the constraints that are proved in the present paper are only valid under additional restrictions on the correlators and computational challenges seem to have so far prevented a generalisation.

We thank the Referee for their time and effort invested in reading and evaluating our manuscript. We are glad to see that they recognize the interest of our work.

Motivated by the comments of the Referee, we could now prove the general NSI inequality (eq. 10), the validity of which was only conjectured in the previous version of the manuscript. We provide a fully analytical proof. Thus we obtain now a general NSI constraint, which can be applied to any probability distribution, lifting the need of any additional restriction on the correlators. We hope this will convince the Referee that our manuscript is worth publishing in Nature Communications. Patrick Remy, a colleague who helped us to obtain this proof was added to the list of authors.

More generally, we agree with the Referee that previous works have discussed theory-independent limits to correlations in networks, as we acknowledge in the introduction of our paper (the list of which has been updated, thanks to the comments of the Referee). Nevertheless, we believe that our work goes well beyond these initial works. Our approach leads to general and strong NSI constraints, some of which are in fact tight as we show. Our approach also identifies the mechanism that would lead to signaling, which thus provides a clear physical meaning to these bounds. Our results indicate the possibility of having nonlocality in the simplest triangle network of binary outputs. Finally, our method is in principle directly applicable to more general networks.

- *The paper is nicely written and I am convinced of its correctness. I have several specific comments, where I think the manuscript could be improved.*

1) All inequalities that are derived in the present paper are restricted to special cases. A comment on the robustness of these in cases where the marginals are only approximately uniform or the symmetries only approximately met would therefore be relevant.

As mentioned above, the new version of the manuscript now includes a general NSI constraint, which can be applied to any statistics.

- *2) I am a bit confused by the claim 'Here, for the first time, we derive explicitly bounds that all correlations satisfying the NSI constraints, whether quantum or post-quantum, have to satisfy.' at the end of page 3. It is mentioned in Section 5.4 of [23] that inflations without inflationary fan-outs apply to all GPTs. These include the so-called Cut Inflation, for which Example 4 of [23] provides an inequality for correlators. A clarification why this is not considered to be such an example and a comparison of the inequalities of the present manuscript to that of Example 4 of [23] would be helpful here.*

We agree with the Referee that our previous terminology was not adequate and confusing. We have corrected this in the new version of the manuscript, and have added a comparison to the results of Ref. [23]. We thank the Referee for pointing out this particular example in [23] which had escaped our attention.

The so-called cut inflation presented in [23] is indeed an inflation of the triangle network which is applicable to non-signalling resources. It is worth noting that this scenario does not include the correlator E_{AC} , and is therefore bound to provide no constraint on this term, contrary to our approach. Nevertheless, an inequality was given in Example 4 of [23], which is also applicable to our scenario. In the E_1 - E_2 plane, this inequality amounts to $2E_2 \leq 1 + E_1^2$. We added now the corresponding curve on Fig. 2 of the main text for comparison. One can see that this inequality is

weaker compared to the one we present, which is in fact tight as it can be saturated by a trilocal model.

- 3) *There is further recent work that proposed theory-independent constraints for networks (I know of [23], [30], [Chaves and Budroni, PRL 116, 240501, 2016] and [Weilenmann and Colbeck, ArXiv:1812.04327]). A comment on the advantages of the present work over these attempts would be of interest to the community.*

We thank the Referees for bringing these relevant references to our attention. We now mention these works in the introduction of the paper. As mentioned above, we have added a comparison the constraint presented in Ref. [23], see Fig. 2. We also now mention the entropic condition of Ref. [30]. Such conditions are however usually quite weak due to the fact that entropy is a coarse-graining of the statistics, which notably does no longer distinguish between correlations and anti-correlations. Finally, the works of Chaves & Budroni and Weilenmann & Colbeck present entropic inequalities for different causal structures than the one considered here, such as bilocality.

- 4) *There seem to be two typos concerning the referencing. [15] and [16] seem to refer to the same paper. In the Supplementary Information, some of the references point to equations rather than figures.*

We thank the Referee for their careful reading of our manuscript. We have checked and corrected these points wherever we noticed a typo.

Reviewers' Comments:

Reviewer #1:

Remarks to the Author:

Dear authors

Thank you for taking such good care of addressing the issues I had raised in my referee report for the original submission. Having re-read the main text with renewed pleasure (without checking the math, which added to the pleasure!), I am happy to recommend enthusiastically acceptance of your revised manuscript in Nature Communications.

Being a perfectionist maniac, however, I still have a few suggestions for minor improvements in both the main text and the supplementary material (the latter of which I did not re-read carefully). Please refer to the notes I inserted on highlighted text in the enclosed document 230009_1_main_and_supp-Referee1.pdf. Although I hope you will not, you may safely ignore all my suggestions, except for a couple typographical errors I noticed in the supplementary material (notive and relabbling) as well as two grammatical errors in the main text (the word "plethora" must be preceded by a determiner such as "a", and "a NSI" should be "an NSI"). Ah yes, the first letter of sentences should always be capitalized.

Stay healthy.

- Gilles Brassard FRS

Reviewer #2:

Remarks to the Author:

In the new version the authors have successfully addressed all my concerns. I recommend publication.

Reviewer #3:

Remarks to the Author:

With the proof of the general inequality (10) and shifting the focus onto binary outcome measurements the manuscript has been considerably improved. Inequality (10) makes it completely clear that the constraints proved here are stronger than the ones known from previous work; the question whether there are non-trilocal distributions when only considering binary outcomes is interesting and still open. I therefore recommend the paper for publication in Nature Communications.

Response to the first Referee

Excerpt 1:

Dear authors

Thank you for taking such good care of addressing the issues I had raised in my referee report for the original submission. Having re-read the main text with renewed pleasure (without checking the math, which added to the pleasure!), I am happy to recommend enthusiastically acceptance of your revised manuscript in Nature Communications.

Being a perfectionist maniac, however, I still have a few suggestions for minor improvements in both the main text and the supplementary material (the latter of which I did not re-read carefully). Please refer to the notes I inserted on highlighted text in the enclosed document 230009_1_main_and_supp-Referee1.pdf. Although I hope you will not, you may safely ignore all my suggestions, except for a couple typographical errors I noticed in the supplementary material (notive and relabbling) as well as two grammatical errors in the main text (the word "plethora" must be preceded by a determiner such as "a", and "a NSI" should be "an NSI"). Ah yes, the first letter of sentences should always be capitalized.

Stay healthy.

- Gilles Brassard FRS

We thank the Referee for his reading of the new version of our manuscript and his renewed recommendation towards acceptance. We have carefully considered all his comments, including the ones in the attached pdf document, and improved the manuscript accordingly.

Response to the second Referee

Excerpt 1:

In the new version the authors have successfully addressed all my concerns. I recommend publication.

We thank the Referee for his/her careful reading of the new version of our manuscript and positive recommendation.

Response to the third Referee

Excerpt 1:

With the proof of the general inequality (10) and shifting the focus onto binary outcome measurements the manuscript has been considerably improved. Inequality (10) makes it completely clear that the constraints proved here are stronger than the ones known from previous work; the question whether there are non-trilocal distributions when only considering binary outcomes is interesting and still open. I therefore recommend the paper for publication in Nature Communications.

We thank the Referee for his/her careful reading of the new version of our manuscript. We are glad to hear that he/she appreciates inequality (10) and recommends publication of our results.